# LIA: Latently Invertible Autoencoder with Adversarial Learning

## Abstract

Deep generative models such as Variational AutoEncoder (VAE) and Generative Adversarial Network (GAN) play an increasingly important role in machine learning and computer vision. However, there are two fundamental issues hindering their real-world applications: the difficulty of conducting variational inference in VAE and the functional absence of encoding real-world samples in GAN. In this paper, we propose a novel algorithm named Latently Invertible Autoencoder (LIA) to address the above two issues in one framework. An invertible network and its inverse mapping are symmetrically embedded in the latent space of VAE. Thus the partial encoder first transforms the input into feature vectors and then the distribution of these feature vectors is reshaped to fit a prior by the invertible network. The decoder proceeds in the reverse order of the encoder's composite mappings. A two-stage stochasticity-free training scheme is designed to train LIA via adversarial learning, in the sense that the decoder of LIA is first trained as a standard GAN with the invertible network and then the partial encoder is learned from an autoencoder by detaching the invertible network from LIA. Experiments conducted on the FFHQ face dataset and three LSUN datasets validate the effectiveness of LIA for inference and generation[1].

## 1    Introduction

Deep generative models play a more and more important role in cracking challenges in computer vision as well as in other disciplines, such as high-quality image generation (Isola et al., 2017; Zhu et al., 2017; Karras et al., 2018a;b; Brock et al., 2018), text-to-speech transformation (van den Oord et al., 2016a; 2017), information retrieval (Wang et al., 2017), 3D rendering (Wu et al., 2016; Eslami et al., 2018), and signal-to-image acquisition (Zhu et al., 2018). Overall, the generative models fall into four categories: autoencoder and its most important variant of Variational Auto-Encoder (VAE) (Kingma & Welling, 2013), auto-regressive models (van den Oord et al., 2016b;a), Generative Adversarial Network (GAN) (Goodfellow et al., 2014), and normalizing flows (NF) (Tabak & Vanden-Eijnden, 2010; Tabak & Turner, 2013; Rezende & Mohamed, 2015).

Here we compare these models through the perspective of data dimensionality reduction and reconstruction. To be formal, let $x$ be a data point in the $d_x$-dimensional observable space $\mathbb{R}^{d_x}$ and $y$ be its corresponding low-dimensional representation in the feature space $\mathbb{R}^{d_y}$. The general formulation of dimensionality reduction is

$$f : \mathbb{R}^{d_x} \to \mathbb{R}^{d_y},\ x \mapsto y = f(x),$$

where $f(\cdot)$ is the mapping function and $d_y \ll d_x$. The manifold learning aims at requiring $f$ under various constraints on $y$ (Tenenbaum1 et al., 2000; Roweis & Saul, 2000). However, the sparsity of data points in high-dimensional space often leads to model overfitting, thus necessitating research on opposite mapping from $y$ to $x$, i.e.

$$g : \mathbb{R}^{d_y} \to \mathbb{R}^{d_x},\ y \mapsto x = g(y),$$

where $g(\cdot)$ is the opposite mapping function with respect to $f(\cdot)$, to reconstruct the data. In general, the role of $g(\cdot)$ is a regularizer to $f(\cdot)$ or a generator to produce more data. The autoencoder is

---

[1]We will make source code of LIA publicly available

of mapping $\boldsymbol{x} \overset{f}{\mapsto} \boldsymbol{y} \overset{g}{\mapsto} \tilde{\boldsymbol{x}}$. A common assumption in autoencoder is that the variables in low-dimensional space are usually sampled from a prior distribution $\mathcal{P}(\boldsymbol{z}; \boldsymbol{\theta})$ such as uniform or Gaussian. To differentiate from $\boldsymbol{y}$, we let $\boldsymbol{z}$ represent the low-dimensional vector following the prior distribution. Thus we can write

$$g : \mathbb{R}^{d_z} \rightarrow \mathbb{R}^{d_x},\ \boldsymbol{z} \mapsto \boldsymbol{x} = g(\boldsymbol{z}),\ \boldsymbol{z} \sim \mathcal{P}(\boldsymbol{z}; \boldsymbol{\theta}).$$

It is crucial to establish such dual maps $\boldsymbol{z} = f(\boldsymbol{x})$ and $\boldsymbol{x} = g(\boldsymbol{z})$. In the parlance of probability, the process of $\boldsymbol{x} \mapsto \boldsymbol{z} = f(\boldsymbol{x})$ is called inference, and the other procedure of $\boldsymbol{z} \mapsto \boldsymbol{x} = g(\boldsymbol{z})$ is called sampling or generation. VAE is capable of carrying out inference and generation in one framework by two collaborative functional modules. However, it is known that in many cases VAEs are only able to generate blurry images due to the imprecise variational inference. To see this, we write the approximation of the marginal log-likelihood

$$\log p(\boldsymbol{x}) = \log \int p(\boldsymbol{x}|\boldsymbol{z})p(\boldsymbol{z})d\boldsymbol{z} \geq -\mathrm{KL}[q(\boldsymbol{z}|\boldsymbol{x})||p(\boldsymbol{z})] + \mathbb{E}_q[\log p(\boldsymbol{x}|\boldsymbol{z})], \tag{1}$$

where $\mathrm{KL}[q(\boldsymbol{z}|\boldsymbol{x})||p(\boldsymbol{z})]$ is the Kullback-Leibler divergence with respect to posterior probability $q(\boldsymbol{z}|\boldsymbol{x})$ and prior $p(\boldsymbol{z})$. This lower-bound log-likelihood usually produces imprecise inference. Furthermore, the posterior collapse frequently occurs when using more sophisticated decoder models (Bowman et al., 2015; Kingma et al., 2016). These two issues greatly limit the generation capability of the VAE. On the other hand, GAN is able to achieve photo-realistic generation results (Karras et al., 2018a;b). However, its critical limitation is the absence of the encoder $f(\boldsymbol{x})$ for carrying inference on real images. Effort has been made on learning an encoder for GAN under the framework of VAE, however the previous two issues of learning VAE still exist. Normalizing flows can perform the exact inference and generation with one architecture by virtue of invertible networks (Kingma & Dhariwal, 2018). But it requires the dimension $d_x$ of the data space to be identical to the dimension $d_z$ of the latent space, thus posing computational issues due to high complexity of learning deep flows and computing the Jacobian matrices.

Inspired by recent success of GANs (Karras et al., 2018a;b) and normalizing flows (Kingma et al., 2016; Kingma & Dhariwal, 2018), we develop a new model called Latently Invertible Autoencoder (LIA). LIA utilizes an invertible network to bridge the encoder and the decoder of VAE in a symmetric manner. We summarize its key advantages as follows:

- The symmetric design of the invertible network brings two benefits. The prior distribution can be exactly fitted from an unfolded feature space, thus significantly easing the inference problem. Besides, since the latent space is detached, the autoencoder can be trained without variational optimization thus there is no approximation here.

- The two-stage adversarial learning decomposes the LIA framework into a Wasserstein GAN (only a prior needed) and a standard autoencoder without stochastic variables. Therefore the training is deterministic[2], implying that the model will be not affected by the posterior collapse when the decoder is more complex or followed by additional losses such as the adversarial loss and the perceptual loss.

- We compare LIA with state-of-the-art generative models on inference and generation/reconstruction. The experimental results on FFHQ and LSUN datasets show the LIA achieves superior performance on inference and generation.

## 2 LATENTLY INVERTIBLE AUTOENCODER

The neural architecture of LIA is designed such that the data distribution can be progressively unrolled from a complex or manifold-valued one to a given simple prior. We now describe the detail as follows.

### 2.1 NEURAL ARCHITECTURE OF LIA

Framework of LIA is based on the classic VAE and the realization of normalizing flow. As shown in Figure 1f, we symmetrically embed an invertible neural network in the latent space of VAE, following the diagram of mapping process as

---

[2]The "deterministic" and "stochasticity-free" in this paper refer to the second-stage training, i.e. the learning of the inference process.

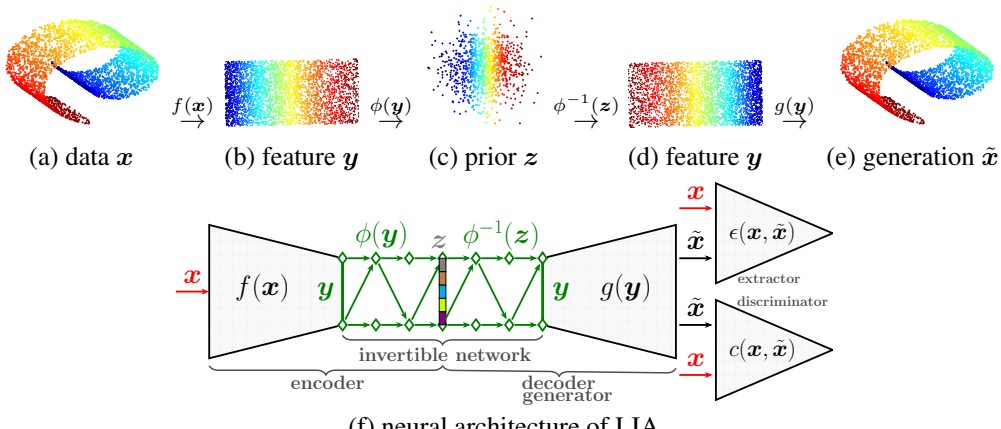

Figure 1: Latently invertible autoencoder (LIA). LIA consists of five functional modules: an encoder to unfold the manifold $\boldsymbol{y} = f(\boldsymbol{x})$, an invertible network $\phi$ to reshape feature embeddings to match the prior distribution $\boldsymbol{z} = \phi(\boldsymbol{x})$ and $\phi^{-1}$ to map latent variables to feature vectors $\boldsymbol{y} = \psi^{-1}(\boldsymbol{z})$, a decoder to produce output $\tilde{\boldsymbol{x}} = g(\tilde{\boldsymbol{y}})$, a feature extractor $\epsilon$ to perform reconstruction measure, and a discriminator $c$ to distinguish real/fake distributions .

$$\overbrace{\boldsymbol{x} \overset{f}{\longmapsto} \boldsymbol{y} \leftrightarrow \underbrace{\boldsymbol{y} \overset{\phi_1}{\mapsto} \boldsymbol{z}_1 \cdots \boldsymbol{z}_{k-1} \overset{\phi_k}{\mapsto} \boldsymbol{z}_k}_{\text{invertible network}}}^{\text{encoder}} \overbrace{\underbrace{\boldsymbol{z}_k \overset{\phi_k^{-1}}{\mapsto} \boldsymbol{z}_{k-1} \cdots \boldsymbol{z}_1 \overset{\phi_1^{-1}}{\mapsto} \boldsymbol{y}}_{\text{invertible network}} \leftrightarrow \boldsymbol{y} \overset{g}{\longmapsto} \tilde{\boldsymbol{x}}}^{\text{decoder}}, \tag{2}$$

where $\phi = \phi_1 \circ \cdots \circ \phi_k$ denotes the deep composite mapping of normalizing flow with depth $k$. LIA first performs nonlinear dimensionality reduction on the input data $\boldsymbol{x}$ and transform them into the low-dimensional feature space $\mathbb{R}^{d_y}$. The role of $f(\boldsymbol{x})$ for LIA can be regarded to unfold the underlying data manifold, as illustrated in Figures 1a and 1b. Therefore, the Euclidean operations such as linear interpolation and vector arithmetic are reliable in this feature space. Then we establish an invertible mapping $\phi(\boldsymbol{y})$ from the feature $\boldsymbol{y}$ to the latent variable $\boldsymbol{z}$, as opposed to the VAEs that directly map original data to latent variables. The feature $\boldsymbol{y}$ can be exactly recovered via the invertibility of $\phi$ from $\boldsymbol{z}$, which is the advantage of using invertible networks. The recovered feature $\boldsymbol{y}$ is then fed into a partial decoder $g(\boldsymbol{y})$ to generate the corresponding data $\tilde{\boldsymbol{x}}$. If the maps $\phi$ and $\phi^{-1}$ of the invertible network are bypassed, it comes back as a standard autoencoder, i.e. $\boldsymbol{x} \overset{f}{\mapsto} \boldsymbol{y} \overset{g}{\mapsto} \tilde{\boldsymbol{x}}$.

In general, any invertible networks are applicable in the LIA framework. We find in practice that a simple invertible network is sufficiently capable of constructing the mapping from the feature space $\mathbb{R}^{d_y}$ to the latent space $\mathbb{R}^{d_z}$. Let $\boldsymbol{x} = [\boldsymbol{x}_t; \boldsymbol{x}_b]$ and $\boldsymbol{z} = [\boldsymbol{z}_t; \boldsymbol{z}_b]$ be the forms of the top and bottom fractions of $\boldsymbol{x}$ and $\boldsymbol{z}$, respectively. Then the invertible network (Dinh et al., 2015) can be built as follows,

$$\boldsymbol{z}_t = \boldsymbol{x}_t, \quad \boldsymbol{z}_b = \boldsymbol{x}_b + \tau(\boldsymbol{x}_t), \tag{3}$$
$$\boldsymbol{x}_t = \boldsymbol{z}_t, \quad \boldsymbol{x}_b = \boldsymbol{z}_b - \tau(\boldsymbol{z}_t), \tag{4}$$

where $\tau$ is the transformation that can be an arbitrary differentiable function. Alternatively, one can attempt to exploit the complex invertible network with affine coupling mappings for more challenging tasks (Dinh et al., 2017; Kingma & Dhariwal, 2018). As conducted in (Dinh et al., 2015), we set $\tau$ as a multi-layer perceptron with the leaky ReLU activation.

## 2.2 RECONSTRUCTION LOSS AND ADVERSARIAL LEARNING

To guarantee the precise reconstruction $\tilde{\boldsymbol{x}}$, the conventional way by (variational) autoencoders is to use the distance $\|\boldsymbol{x} - \tilde{\boldsymbol{x}}\|$ or the cross entropy directly between $\boldsymbol{x}$ and $\tilde{\boldsymbol{x}}$. Here, we utilize the perceptual loss that is proven to be more robust to variations of image details (Johnson et al., 2016). Let $\epsilon$ denote the feature extractor, e.g. VGG (Simonyan & Zisserman, 2014). Then we can write the loss

$$L(\epsilon, \boldsymbol{x}, \tilde{\boldsymbol{x}}) = \|\epsilon(\boldsymbol{x}) - \epsilon(\tilde{\boldsymbol{x}})\|_2. \tag{5}$$

It suffices to emphasize that the functionality of $\epsilon$ here is in essence to produce the representations of the input $\boldsymbol{x}$ and the output $\tilde{\boldsymbol{x}}$. The acquisition of $\epsilon$ is fairly flexible. It can be attained by supervised *or*

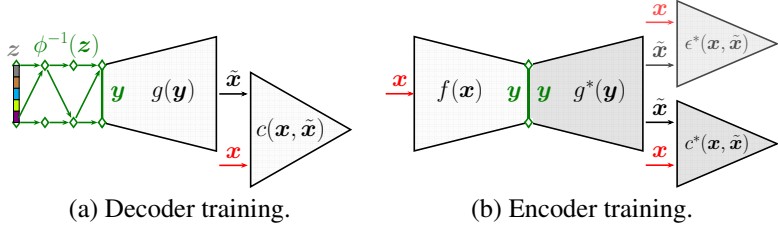

(a) Decoder training.  (b) Encoder training.

Figure 2: Detaching the functional modules of LIA for two-stage training.

unsupervised learning, meaning that $\epsilon$ can be trained with class labels or without class labels (van den Oord et al., 2018).

The norm-based reconstruction constraints usually incur the blurry image generation in autoencoder-like architectures. This problem can be handled via the adversarial learning (Goodfellow et al., 2014). To do so, a discriminator $c$ is employed to balance the loss of the comparison between $\boldsymbol{x}$ and $\tilde{\boldsymbol{x}}$. Using the Wasserstein GAN (Arjovsky et al., 2017; Gulrajani et al., 2017), we can write the optimization objective as

$$L(c, \boldsymbol{x}, \tilde{\boldsymbol{x}}) = \mathop{\mathbb{E}}_{\tilde{\boldsymbol{x}} \sim \mathbb{P}_{\tilde{x}}} [c(\tilde{\boldsymbol{x}})] - \mathop{\mathbb{E}}_{\boldsymbol{x} \sim \mathbb{P}_x} [c(\boldsymbol{x})] + \frac{\gamma}{2} \mathop{\mathbb{E}}_{\boldsymbol{x} \sim \mathbb{P}_x} \left[ \| \nabla_{\boldsymbol{x}} c(\boldsymbol{x}) \|_{\ell_2}^2 \right], \tag{6}$$

where $\mathbb{P}_x$ and $\mathbb{P}_{\tilde{x}}$ denote the probability distributions of the real data and the generated data, respectively. $\gamma$ is the hyper-parameter of the regularization. The $R_1$ regularizer is formulated in (Mescheder et al., 2018), which is proven more stable for training convergence.

## 3 TWO-STAGE STOCHASTICITY-FREE TRAINING

Training deep models usually follows an end-to-end fashion for the whole architecture. To backprop-agate gradients through random variables for VAE, the reparameterization trick is harnessed (Kingma & Welling, 2013), i.e. $\boldsymbol{z} = \boldsymbol{\mu} + \boldsymbol{\sigma} * \mathcal{N}(\mathbf{0}, \mathbf{1})$, where $\boldsymbol{\mu}$ is the mean and $\boldsymbol{\sigma}$ the standard deviation. The regularization of coupling the prior and the posterior is the KL divergence used to optimize the parameters of the encoder by backpropagation. For our framework, however, we find that this end-to-end learning strategy cannot lead the algorithm to converge to a satisfactory optima. To proceed, we propose a scheme of two-stage *stochasticity-free* training, which decomposes the framework into two parts that can be well trained end-to-end respectively, as shown in Figure 2. At the first step, the decoder of LIA is trained using adversarial learning with invertible network to acquire the ability of high-quality generation. At the second step, the invertible network that connects feature space and latent space is detached from the architecture, reducing the framework to a standard autoencoder *without* variational inference. Thus this two-stage design prevents the posterior collapse issue while facilitates the decoder with more complex structure as well as adversarial loss for high-resolution image generation.

### 3.1 DECODER TRAINING

ProGAN (Karras et al., 2018a), StyleGAN (Karras et al., 2018b), and BigGAN (Brock et al., 2018) are capable of generating photo-realistic images from random noise sampled from some prior distribution. Then it is naturally supposed that such GAN models are applicable to recover a precise $\tilde{\boldsymbol{x}}$ if we can find the latent variable $\boldsymbol{z}$ for the given $\boldsymbol{x}$. Namely, we may train the associated GAN model separately in the LIA framework. To conduct this, we single out a standard GAN model for the first-stage training, as displayed in Figure 2a. Here $\boldsymbol{z}$ is directly sampled from a prior distribution. According to the principle of Wasserstein GAN, the optimization objective can be written as

$$\{\phi^*, g^*, c^*\} = \min_{\phi, g} \max_c L(c, \boldsymbol{x}, \tilde{\boldsymbol{x}}), \tag{7}$$

where the superscript $*$ denotes that the parameters of corresponding mappings have already been learned. It is worth noting that the role of the invertible network here is just its transformation invertibility. We do not pose any constraints on the probabilities of $\boldsymbol{z}$ and $\phi(\boldsymbol{y})$ in contrast to normalizing flows. Our strategy of attaching an invertible network in front of the generator can be potentially applied to any GAN models.

## 3.2 ENCODER TRAINING

In the LIA architecture, the invertible network is embedded in the latent space in a symmetric fashion, in the sense that $f(\boldsymbol{x}) = \boldsymbol{y} = \phi^{-1}(\boldsymbol{z})$. The unique characteristic of the invertible network allows us to detach the invertible network $\phi$ from the LIA framework. Thus we attain a conventional autoencoder without stochastic variables, as shown in Figure 2b. In practice, the feature extractor $\epsilon$ in perceptual loss is the VGG weight up to conv4 pretrained on the ImageNet dataset. After the first-stage encoder training, the parameter of $f$ is needed to be learned as

$$f^* = \min_f \beta L(\epsilon^*, \boldsymbol{x}, \tilde{\boldsymbol{x}}) + L(c^*, \boldsymbol{x}, \tilde{\boldsymbol{x}}), \tag{8}$$

where $\beta$ is the hyper-parameter. The above optimization serving to the architecture in Figure 2b is widely applied in computer vision. It is the backbone framework of various GANs for diverse image processing tasks (Isola et al., 2017; Zhu et al., 2017). For LIA, however, it is much simpler because we only need to learn the partial encoder $f$. This simplicity brought by the two-stage training is able to enforce the encoder to converge with more precise inference.

## 4 RELATED WORK

Our LIA model is relevant to the works that solve the inference problem for VAEs with adversarial learning as well as the works that design encoders for GANs. The integration of GAN with VAE can be traced back to the work of VAE/GAN (Larsen et al., 2016) and implicit autoencoders (Makhzani et al., 2015; Makhani, 2018). These methods encounter the difficulty of end-to-end training, because the gradients are prone to becoming unstable after going through the latent space in deep complex architectures (Bowman et al., 2015; Kingma et al., 2016). Besides, there is an intriguing attempt of training VAE in the adversarial manner (Ulyanov et al., 2017; Heljakka et al., 2018). These approaches confront the trade-off between the roles of the encoder that performs inference and compares the real/fake distributions. This is difficult to tune. So we prefer the complete GAN with an indispensable discriminator.

The closely related works to LIA are the models of combining VAE and the inverse autoregressive flow (Kingma et al., 2016) and the latent-flow-based VAE approach that are VAEs with latent variables conditioned by normalizing flows (Su & Wu, 2018; Xiao et al., 2019). These three models all need to optimize the posterior probability of normalizing flows, which is essentially different from our deterministic optimization in LIA. The stochasticity-free training is directly derived from the symmetric design of the invertible network in the latent space, which is different from (Kingma et al., 2016) and (Su & Wu, 2018; Xiao et al., 2019). There are alternative attempts of specifying the generator of GAN with normalizing flow (Grover et al., 2017) or mapping images into feature space with partially invertible network (Lucas et al., 2019). These approach suffers from high complexity computation for high dimensions. The approach of two-stage training in (Luo et al., 2017) is incapable of solving the posterior estimation.

It worth noticing that the reconstruction task we focus here is different to the recent work of representation learning which learns features for recognition using adversarial inference (Dumoulin et al., 2017; Jeff Donahue, 2017; Donahue & Simonyan, 2019). Our primary goal is to faithfully reconstruct real images from the latent code.

## 5 EXPERIMENTS

For experimental setup, we instantiate the decoder of LIA with the generator of StyleGAN (Karras et al., 2018b). The difference is that we replace the mapping network (MLP) in StyleGAN with the invertible network. The layer number of the invertible network is 8. The hyper-parameters for the discriminator are $\gamma = 10$ (equation (6)) and $\beta = 0.001$ (equation (8)). For perceptual loss in equation (5), we take $\epsilon = \text{conv4\_3}$ from the VGG weight.

The generative models we compare are the MSE-based optimization methods (Radford et al., 2016; Berthelot et al., 2017; Lipton & Tripathi, 2017)[3], the adversarially learned inference (ALI) (Dumoulin et al., 2017), and the adversarial generator-encoder (AGE) network (Ulyanov et al., 2017). To evaluate

---

[3]We use the code at `https://github.com/Puzer/stylegan-encoder`.

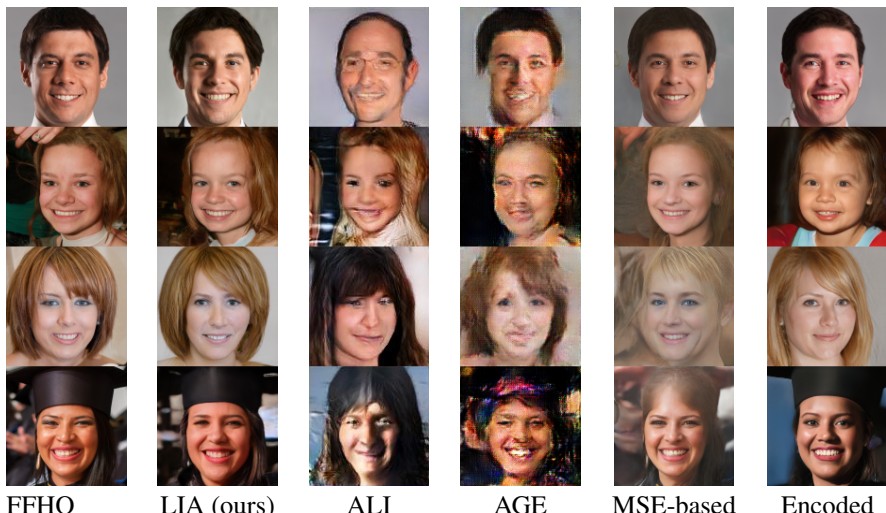

FFHQ     LIA (ours)     ALI     AGE     MSE-based     Encoded

Figure 3: Reconstructed faces by generative models on FFHQ database.

Table 1: Quantitative comparison of image reconstruction.

| Metric | LIA (ours) | ALI | AGE | MSE-based optimization | Encoded StyleGAN |
|--------|------------|-------|--------|------------------------|------------------|
| FID | **16.88** | 74.98 | 118.88 | 44.79 | 22.26 |
| SWD | **10.01** | 15.09 | 38.70 | 43.44 | 15.82 |
| MSE | **18.10** | 32.61 | 29.91 | 18.81 | 23.18 |

the necessity of the invertible network, we also train an encoder and a StyleGAN with its original multi-layer perceptron, which is the last column in Figure 3. The two-stage training scheme is used as LIA does. The generator and discriminator of the StyleGAN is exactly same to that of StyleGAN.

For quantitative evaluation metrics, we use Fréchet inception distance (FID), sliced Wasserstein distance (SWD), and mean square error (MSE). These three metrics are commonly used to measure the numerical accuracy of generative algorithms (Ulyanov et al., 2017; Karras et al., 2018a; Jeff Donahue, 2017; Karras et al., 2018b). We directly use the code released by the authors of ProGAN (Karras et al., 2018a). The prior for $z$ is Gaussian and the dimension is 512.

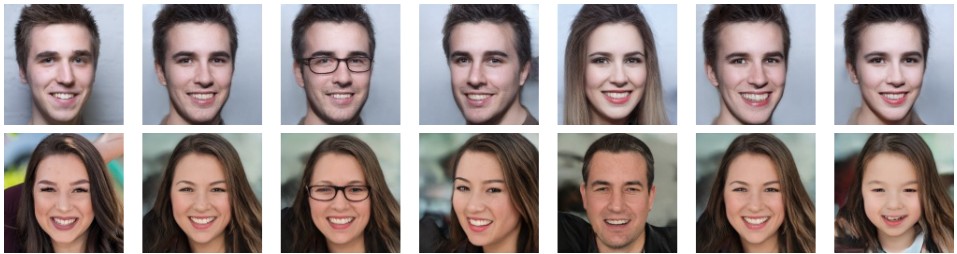

Figure 4: Manipulating reconstructed faces. The algorithm in (Shen et al., 2019) is applied to manipulate the image reconstructed from the latent code $w$ given by LIA. Each row shows the original image, the reconstruction, glass, pose, gender, smile, and age.

## 5.1 FFHQ FACE DATABASE

All models are first tested on the Flickr-Faces-HQ (FFHQ) database[4] created by the authors of StyleGAN as the benchmark. FFHQ contains 70,000 high-quality face images. We take the first 65,000 faces as the training set and the remaining 5,000 faces as the reconstruction test according to the exact order of the dataset. We do not split the dataset by random sampling for interested readers can precisely reproduce all the reported results with our experimental protocol.

---

[4] https://github.com/NVlabs/ffhq-dataset

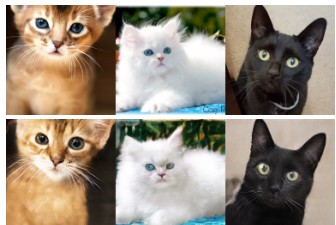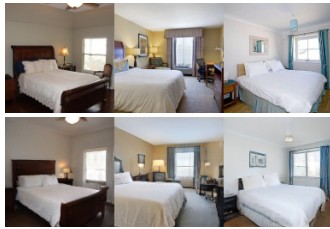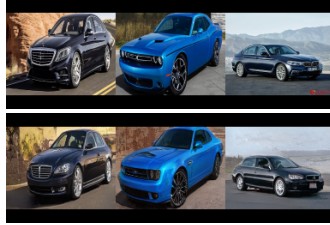

Figure 5: The exemplar real images of objects and scenes from LSUN database and their reconstructed images by LIA. Three categories are tested, *i.e.* cat, bedroom, and car. For each group, the first row is the original images, the second row shows the reconstructed images.

Figure 3 shows the reconstructed faces of all models. It is clear that LIA significantly outperforms others. The reconstructed faces by ALI and AGE look correct, but the quality is mediocre. The ideas of ALI and AGE are elegant. Their performance may be improved with the new techniques such as progressive growing of neural architecture or style-based one. The method of the MSE-based optimization produces facial parts of comparable quality with LIA when the faces are normal. But this approach fails when the variations of faces become large. For example, the failure comes from the long fair, hats, beard, and large pose. The interesting phenomenon is that the StyleGAN with encoder only does not succeed in recovering the target faces using the same training strategy as LIA, even though it is capable of generating photo-realistic faces in high quality due to the StyleGAN generator. This indicates that the invertible network plays the crucial role to make the LIA work. The quantitative result in Table 1 shows the consistent superiority of LIA. More reconstruction results are included in Appendix A.5. The reconstruction from LIA facilitates semantic photo editing. Figure 4 shows the manipulation results on reconstructed faces. More results can be found in Appendix A.7. The interpolation and style mixing are also displayed for reference in Appendix A.2.

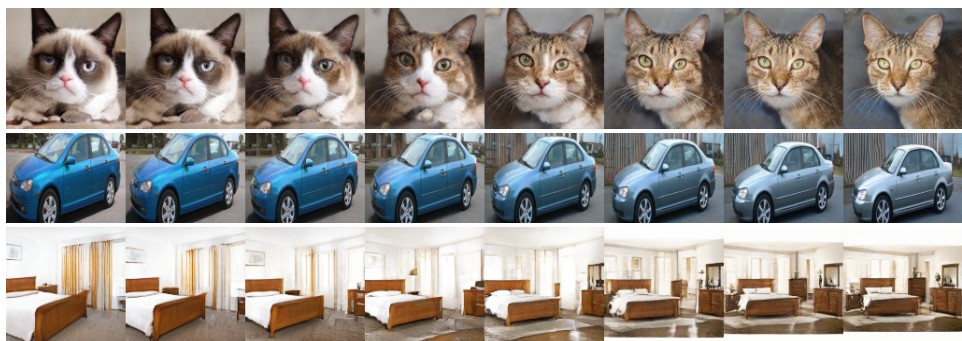

Figure 6: Interpolation on objects in the LSUN database.

## 5.2 LSUN DATABASE

To further evaluate LIA on the data with large variations, we use the three categories from the large-scale LSUN database (Yu et al., 2015), i.e. cat, car, and bedroom. For each category, the 0.1 million images are selected by ranking algorithm (Zhou et al., 2003) from the first 0.5 million images in the dataset. Each cat and bedroom images are resized to be $128 \times 128$ and the size of the car image is $128 \times 96$ for training. We take subsets because it does not take too long for training to converge while still maintains the data complexity. These subsets will be made available for evaluation.

Figure 5 shows that the reconstructed objects by LIA faithfully maintain the semantics as well as the appearance of the original ones. For example, the cats' whiskers are recovered, indicating that LIA is able to recover very detailed information. More results are attached in Appendix A.6. We can see that LIA significantly improves the reconstruction quality. The improvement mainly comes from the two-stage training of LIA. The decoder trained with adversarial learning guarantees that the generated images are photo-realistic. The encoder deterministically trained with perceptual and adversarial losses ensures that latent feature vectors can be obtained more precisely. This two-stage training is enabled by the design that the invertible network detachs the encoder and decoder, thus avoiding the optimization of the posterior probability when learning the encoder. Figure 6 shows the

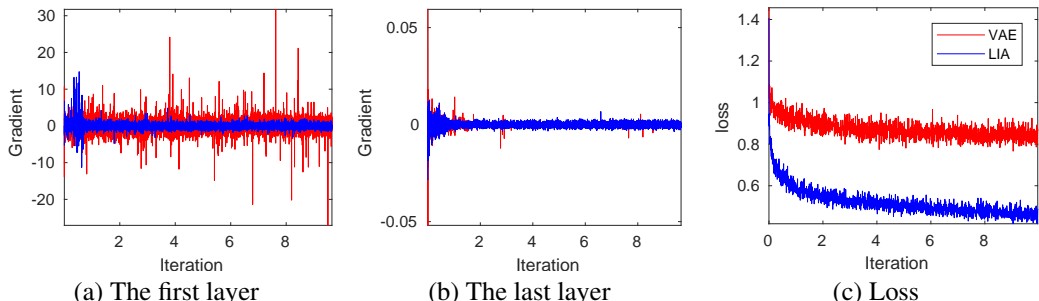

(a) The first layer            (b) The last layer            (c) Loss

Figure 7: Illustration of gradients and losses for training encoders on FFHQ dataset. For VAE, the decoder is exactly the generator of StyleGAN. The two-stage training of VAE is the same as that of LIA. The first and last layers in (a) and (b) refer to the neural architecture of the encoder. The $y$-axis indicates the average value of the gradients. The $x$-axis is re-scaled by 10,000. The gradients for LIA case are much more stable than the ones for VAE. The gradient of middle layer is also shown in Appendix A.8.

interpolation results on those reconstructed objects. More results are included in Appendix A.3. The experimental results on FFHQ and LSUN databases verify that the symmetric design of the invertible network and the two-stage training successfully handles the issue of inference.

### 5.3 WHY THE INVERTIBLE NETWORK MATTERS?

To further reveal the importance of the invertible network in LIA, we investigate the gradients of the encoders under two different cases of decoders when learning the encoder in the two-stage training. The first case is to directly apply the KL-divergence to optimize the random latent space as VAE does. The mapping of the encoder and the decoder is $x \overset{f}{\mapsto} z \overset{\varphi}{\mapsto} y \overset{g}{\mapsto} \tilde{x}$, where $\varphi$ is the mapping network (MLP) in StyleGAN. This is the conventional way of learning encoders for GAN algorithms via variational inference. The second one is our architecture of LIA for the second-stage training, i.e. $x \overset{f}{\mapsto} y \overset{g}{\mapsto} \tilde{x}$, where the latent space is removed due to the symmetric design of the invertible network. Figure 7 clearly illustrates the difference of gradients between these two cases. The gradient volatility for variational inference is high and the associated loss is not effectively reduced, meaning that the gradients during training are noisy and not always informative. This may indicate that the stochasticity in latent space causes problems for training encoder via variational inference. Instead, the encoder's gradients for LIA are rather stable across different layers and the loss decreases monotonically, showing the importance of the stochasticity-free training and the invertible network.

## 6 CONCLUSION

A new generative model, named Latently Invertible Autoencoder (LIA), has been proposed for generating image sample from a probability prior and simultaneously inferring accurate latent code for a given sample. The core idea of LIA is to symmetrically embed an invertible network in an autoencoder. Then the neural architecture is trained with adversarial learning as two decomposed modules. With the design of two-stage training, the decoder can be replaced with any GAN generator for high-resolution image generation. The role of the invertible network is to remove any probability optimization and bridge the prior with unfolded feature vectors. The effectiveness of LIA is validated with experiments of reconstruction (inference and generation) on FFHQ and LSUN datasets.

It is still challenging to faithfully recover *all* the image content especially when the objects or scenes have unusual parts. For example, LIA fails to recover the hand appeared at the top of the little girl (the second row in Figure 3). Besides, the Bombay cat's necklace (the second row in Figure 5) is missed in the reconstructed image. These features belong to multiple unique parts of the objects or scenes, which are difficult for the generative model to capture. One possible solution is to raise the dimension of latent variables (e.g. using multiple latent vectors) or employ the attention mechanism to highlight such unusual structures in the decoder, which is left for future work.

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

# A  APPENDIX

## A.1  GENERATION RESULTS

We compare the generation results of StyleGAN and LIA. For the decoder (generator), the difference between these two models is the mapping network. To be specific, the architecture of StyleGAN generator is of form

$$z \stackrel{\text{MLP}}{\longmapsto} y \stackrel{g}{\longmapsto} \tilde{x}, \tag{9}$$

where MLP is multi-layer perceptron. The LIA decoder consists of mappings

$$z \stackrel{\phi^{-1}}{\longmapsto} y \stackrel{g}{\longmapsto} \tilde{x}, \tag{10}$$

where $\phi$ is the invertible network. We follow the experimental setting of StyleGAN (Karras et al., 2018b) to compute the FIDs. We first generate 50,000 fake faces and then randomly sample another 50,000 real faces. The FID is computed with these two datasets. Table 2 and Figure 8 show that the performance of StyleGAN and LIA is comparably good. The benefit of using the invertible network is to facilitate the training of the encoder for inference.

Table 2: FIDs of generative results.

| Algorithm | FFHQ | Cat | Car | Bedroom |
|-----------|------|------|------|---------|
| StyleGAN | 6.23 | 7.36 | 3.09 | 4.88 |
| LIA | 6.28 | 7.08 | 2.52 | 4.36 |

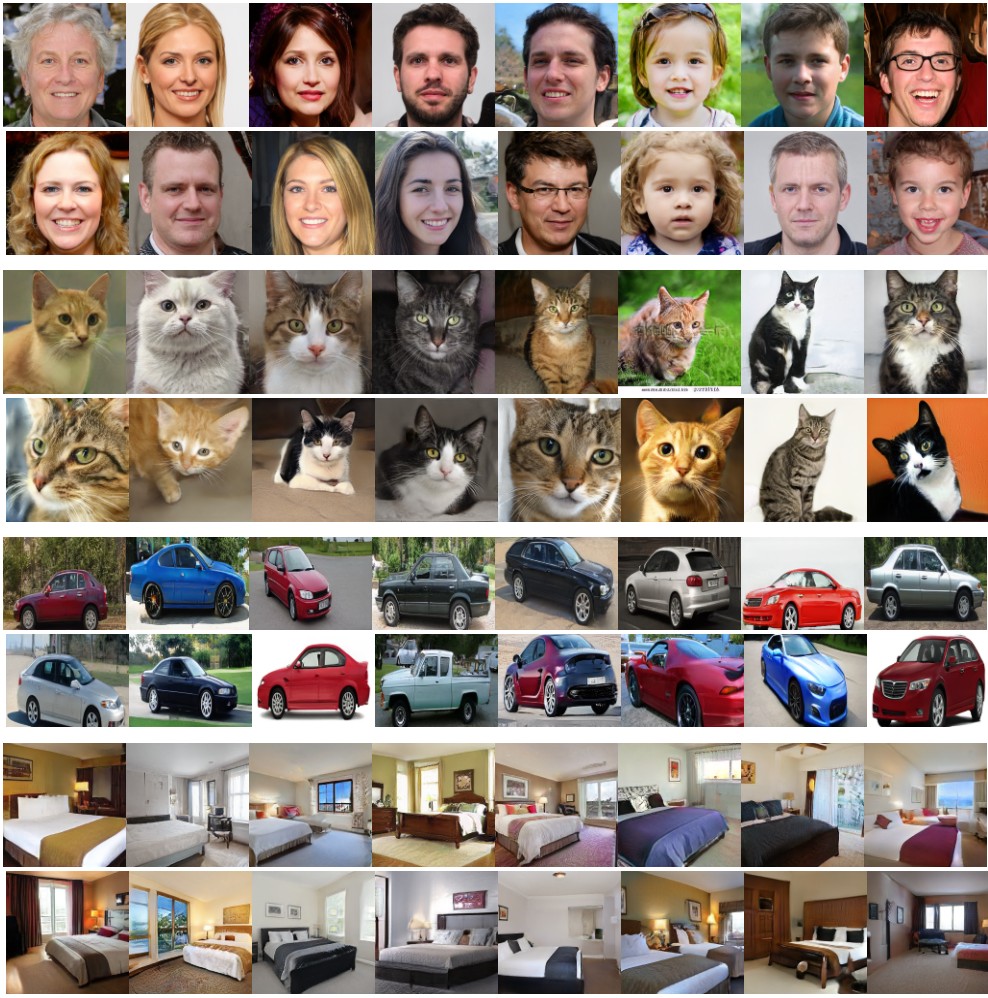

Figure 8: Generative results by StyleGAN and LIA. For each group of objects, the first row shows the generative results obtained by StyleGAN and the second row by LIA.

### A.2 Face Interpolation and style mixing

Examining the interpolation in the latent feature space is an effective way of visualizing the capability of generative models as well as measuring how well it fits the underlying data distribution. Here we compare the three algorithms. As shown in Figure 9, LIA achieves smoother interpolation result while well preserve the facial properties. The interpolation quality of the MSE-based optimization is actually based on the reconstruction performance because it has a good generator (StyleGAN). The intermediate interpolation result from Glow deviates from real faces.

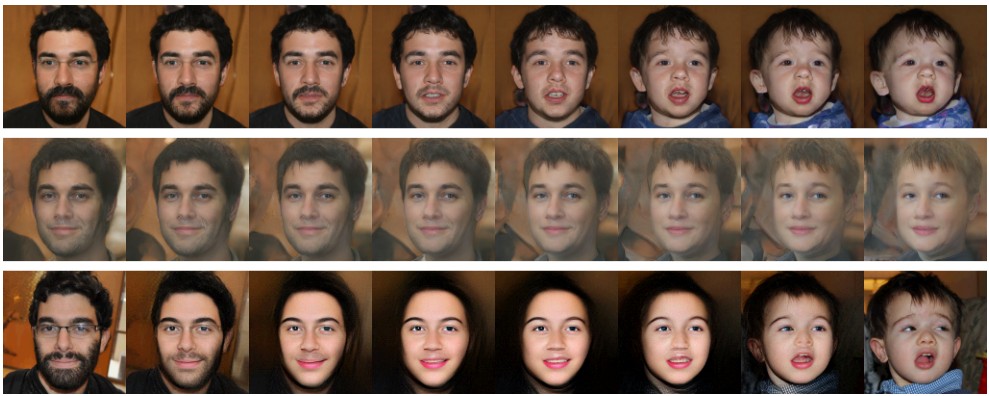

Figure 9: Interpolation results by three generative models. The first row shows the result of LIA (ours), the second row that of the MSE-based optimization, and the last row that of Glow. The first and last faces in the Glow result are the real face images of FFHQ.

We further perform style mixing using a small set of reconstructed faces. Style mixing is conducted using the same approach presented in (Karras et al., 2018b). The difference is that LIA uses the real faces thanks to its encoding capability. Figure 10 shows that our algorithm can infer accurate latent codes and generate high-quality mixed faces.

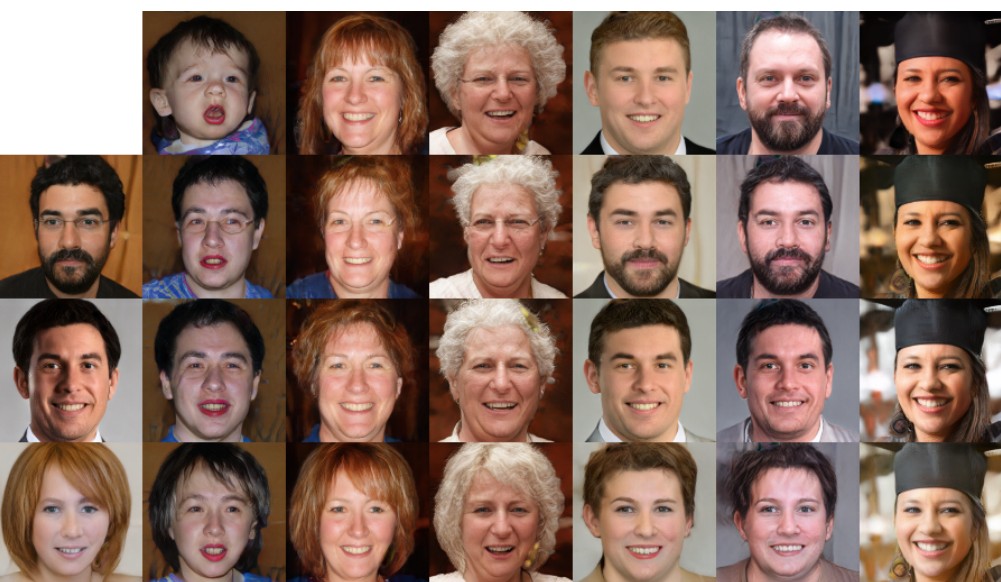

Figure 10: Style mixing for real faces. The faces in the first row are the source faces where the style comes from.

### A.3 OBJECT INTERPOLATION

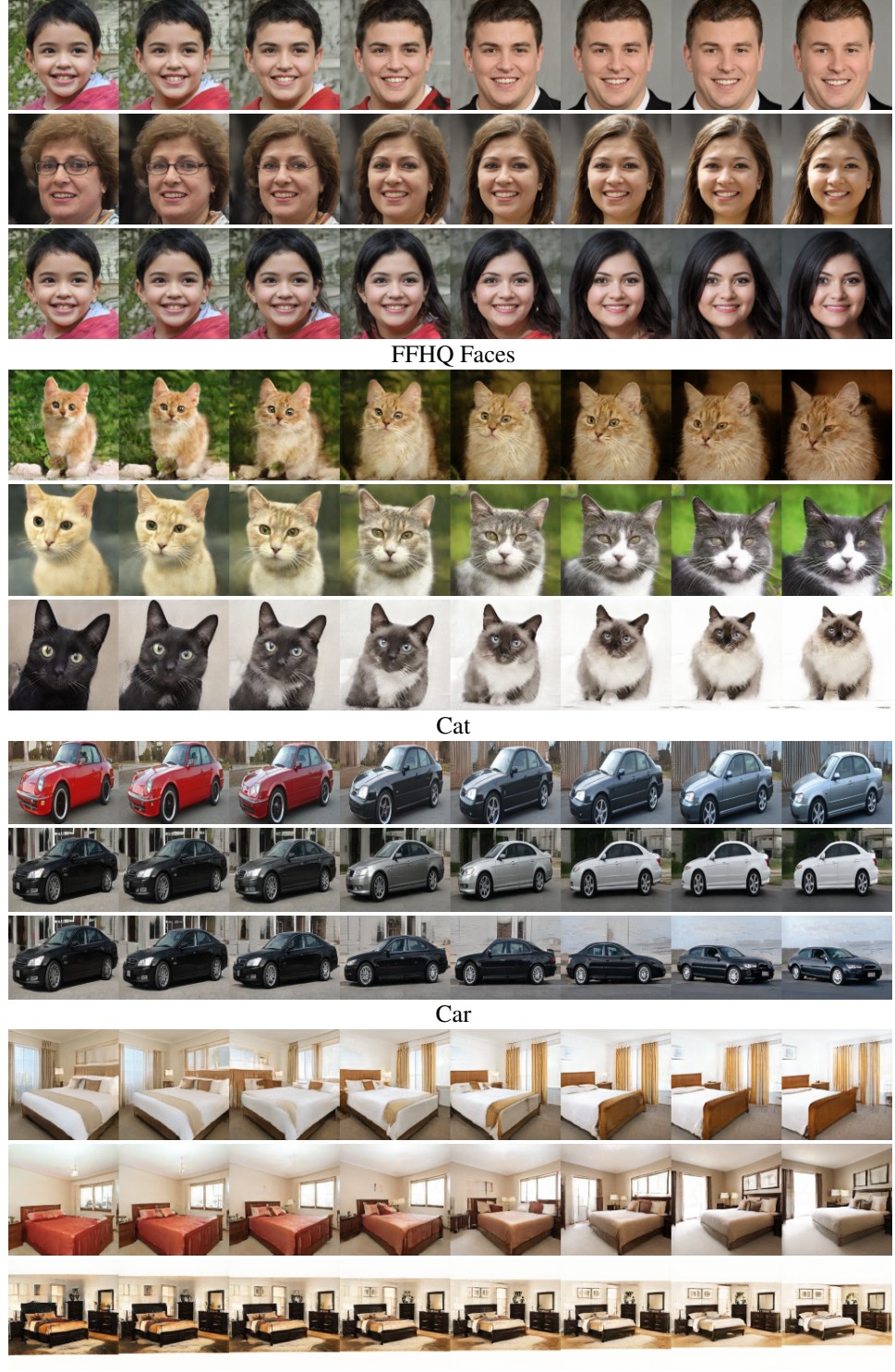

FFHQ Faces

Cat

Car

Bedroom

Figure 11: Interpolation on FFHQ faces and objects in the LSUN database.

## A.4    DISENTANGLEMENT ILLUSTRATION

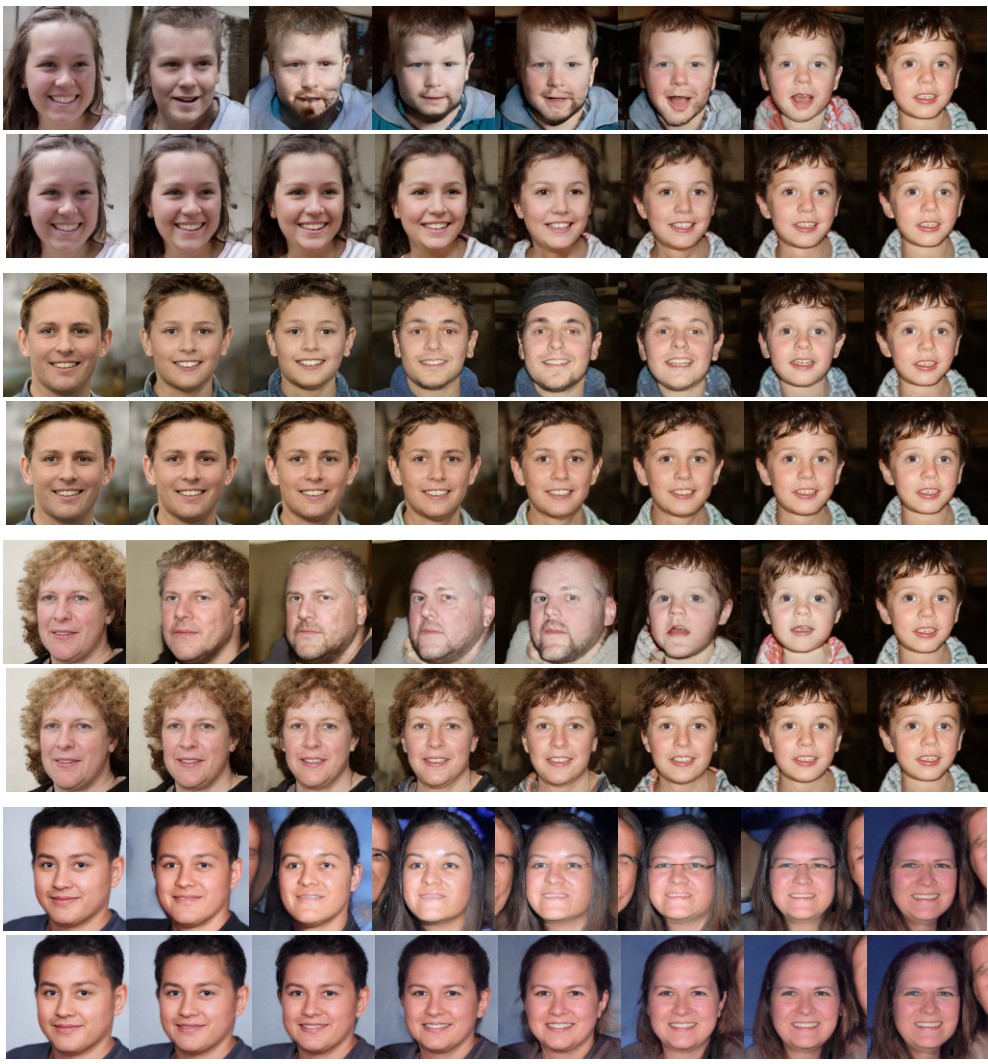

Figure 12: Interpolation with the latent code $z$ and the feature $w$. For each group, the first row shows the result with $z$ and the second row shows the result with $w$.

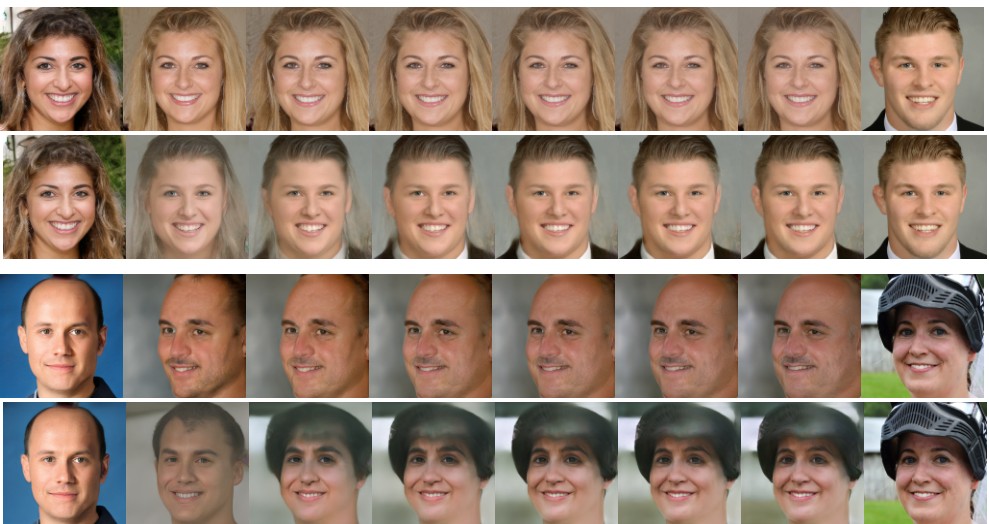

Figure 13: Generated faces along the optimization path with the latent code $z$ and the feature $w$. The path is formed by the optimization of the MSE loss with respect to $z$ and $w$, respectively. The faces in the first column are generated by the initial values of $z$ and $w$. Then the following faces are sequentially generated with the optimization process until the optimization converges. For each group, the first row shows the result with $z$ and the second row shows the result with $w$. The disentanglement effect of $w$ is clearly shown. Instead, the latent code $z$ suffers from the entanglement of features.

## A.5 RECONSTRUCTION FOR FFHQ DATABASE

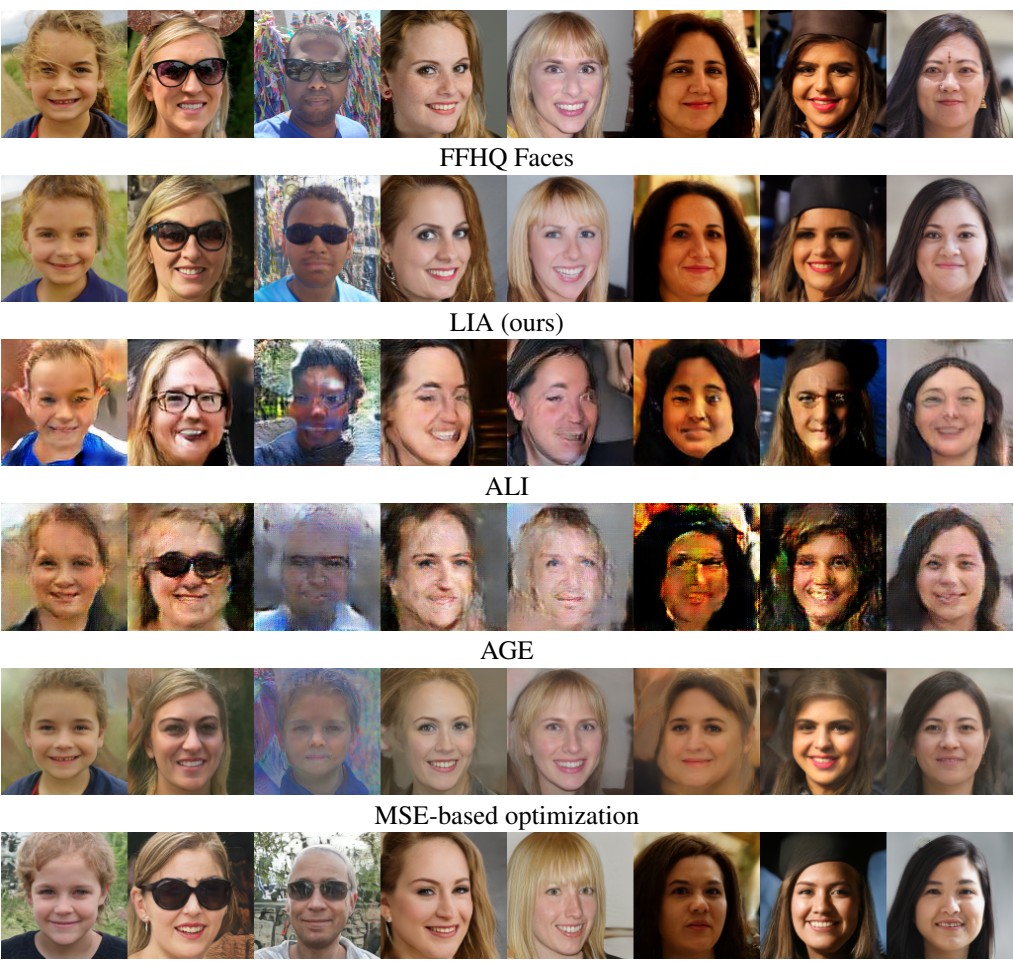

Figure 14: Real images from FFHQ dataset and their reconstructed faces by various generative models.

## A.6   RECONSTRUCTION FOR LSUN DATABASE

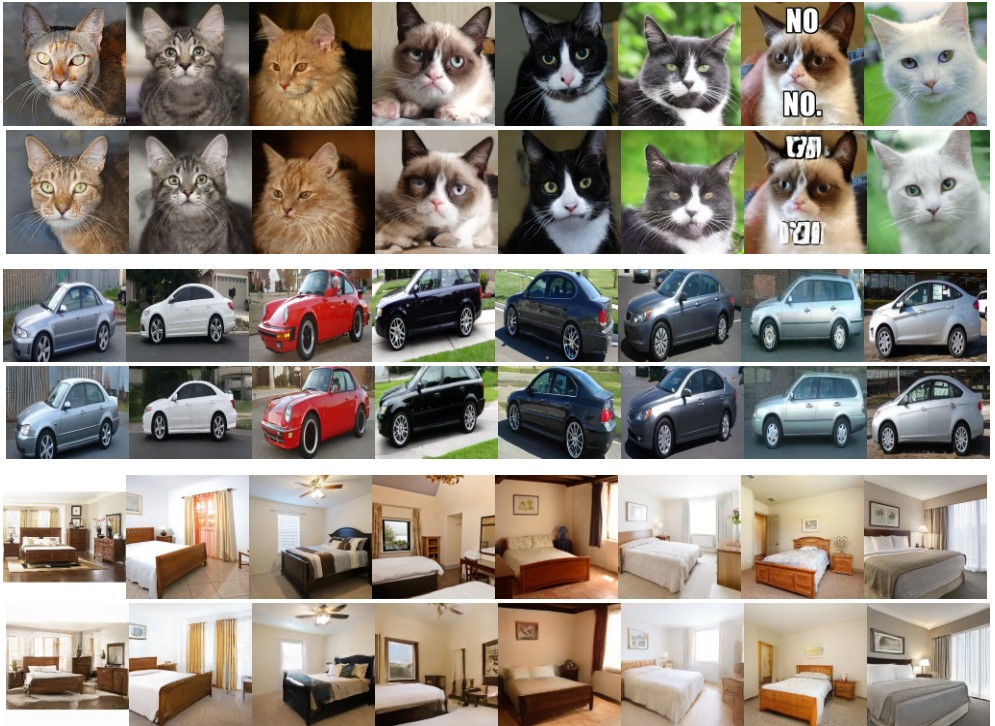

Figure 15: Real images from LSUN dataset and the reconstructed images by LIA.

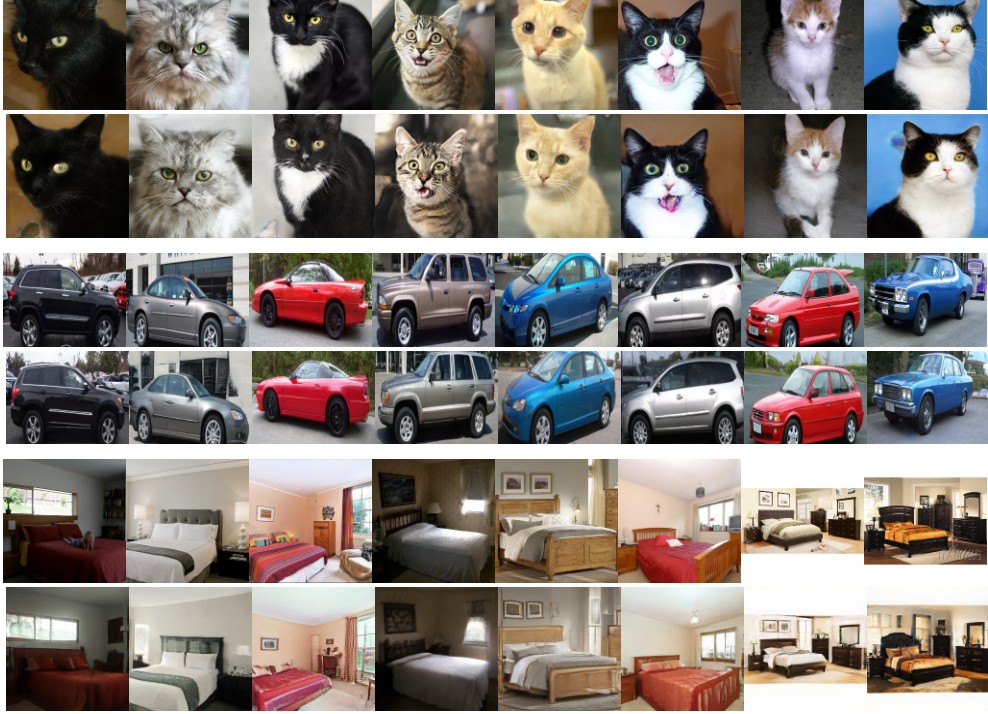

Figure 16: Real images from LSUN dataset and the reconstructed images by LIA.

## A.7 FACE MANIPULATION

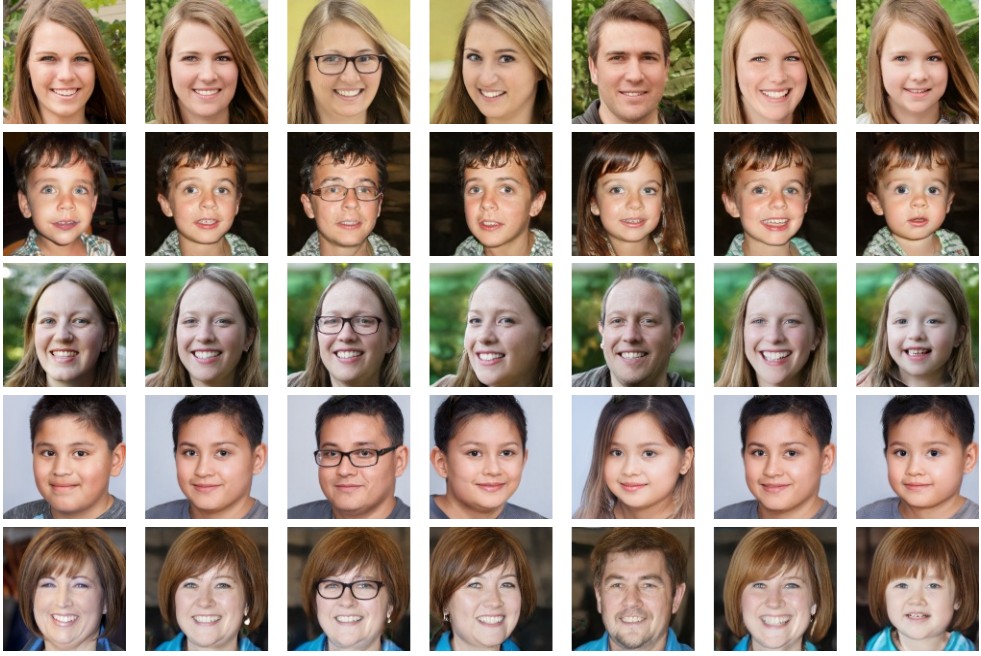

Figure 17: Manipulating reconstructed faces. The algorithm in (Shen et al., 2019) is applied to manipulate faces reconstructed from the latent code $w$ given by LIA. Each row shows the original image, the reconstruction, glass, pose, gender, smile, and age.

### A.8 GRADIENTS OF MIDDLE LAYER

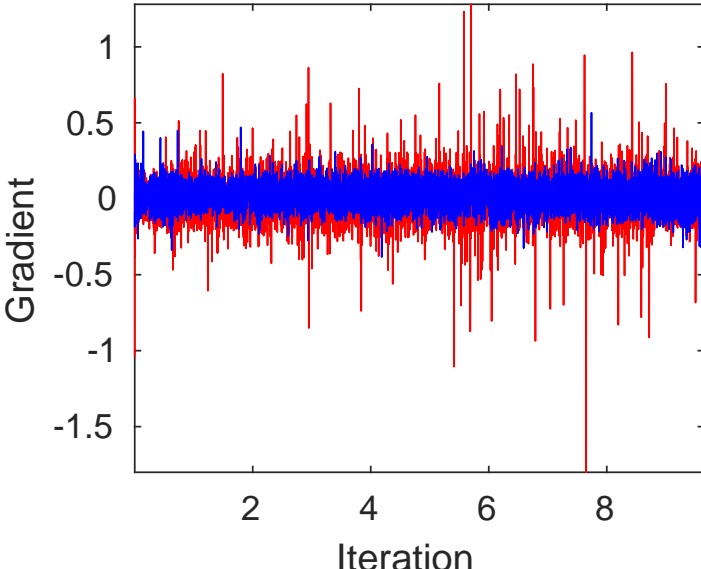

Figure 18: Illustration of gradients when training encoders on FFHQ dataset. For VAE, the decoder is exactly the generator of StyleGAN. The two-stage training of VAE is the same as that of LIA. The red and blue lines represent the gradients for VAE and LIA respectively. This figure shows the gradient of middle layer in the neural architecture of the encoder, which is complementary to Figure 7.

## A.9 THE ARCHITECTURAL DETAIL OF LIA

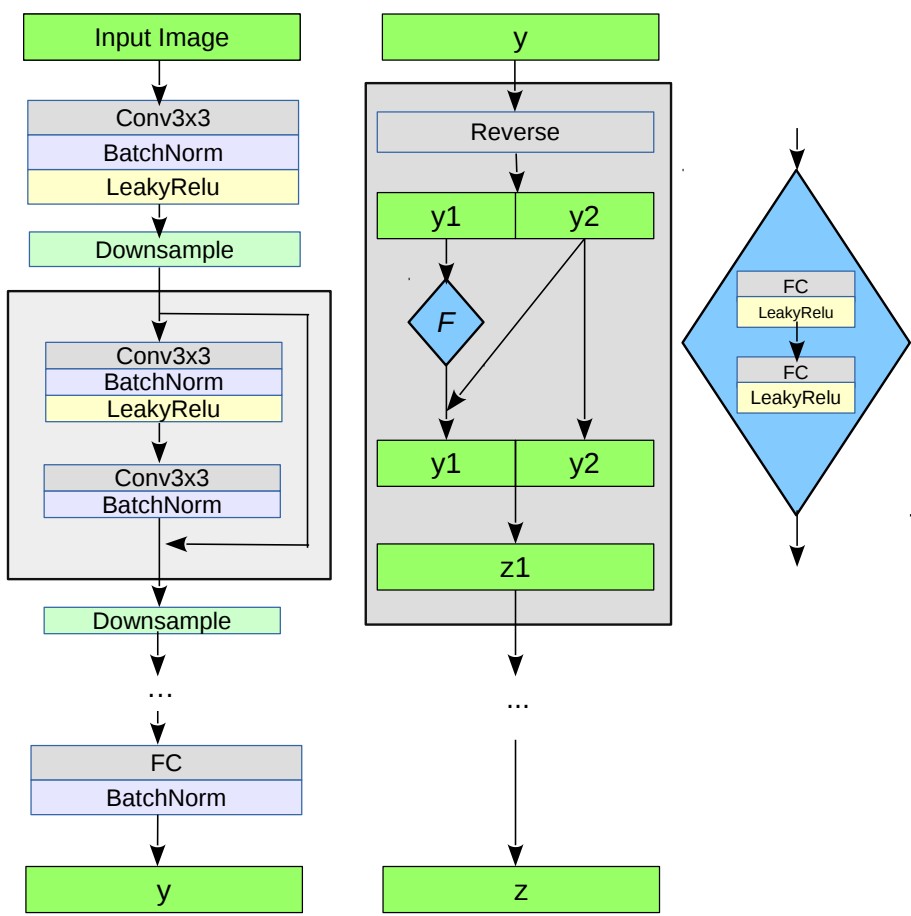

Figure 19: The architectural detail of LIA for each block of the whole network. Here the encoder and the invertible network are shown. The synthesis network in the decoder is the same as StyleGAN.

