# OpenReview forum: "LIA: Latently Invertible Autoencoder with Adversarial Learning"
_ICLR.cc/2020/Conference — Reject_

### Official Review · AnonReviewer1 · 2019-10-20
**Official Blind Review #1**

**Rating:** 3

**Review:**

LIA: Latently Invertible Autoencoder Review

This paper proposes a novel generative autoencoder, and a two-stage scheme for training it. A typical VAE is trained with a variational approximation: during training latents are sampled from mu(x) + sigma(x) * N(0,1), mu and sigma are regularized with KL div to match an isotropic normal, and the model minimizes a reconstruction loss. A LIA is instead first trained as a standard GAN, where an invertible model, Phi, (e.g. a normalizing flow) is hooked up to the Generator/Decoder, such that the output is G(Phi^{-1}(z)) with z~p(z), a simple distribution. In the second stage, the Generator/Decoder and Discriminator are frozen, and an Encoder is trained to minimize a reconstruction loss (in this paper, a pre-trained VGG network is used as a feature extractor to produce a perceptual loss) and maximize the “real” output of the frozen Discriminator on reconstructed samples.

The key advantage of this method is that during the second stage no stochasticity is injected into the network and Phi is not involved.This means that the encoder does not need to be regularized to produce a specific parametric form of its output (no KL(p || q))), instead implicitly learning to match the latent distribution expected by the generator through the reconstruction losses. Additionally, because the latent space is not e.g. an isotropic Gaussian, it can be more expressive and flexible, only being constrained to an invertible transformation of the distribution p(z) chosen during the first training stage.

The training procedure is evaluated using a StyleGAN architecture on several high-res, single-class datasets (faces and LSUN Cats/Cars/Bedrooms). The quality of the resulting reconstructions is compared against several methods which are also capable of inference (like ALI and post-training an encoder on a standard StyleGAN), and samples and interpolations are presented. There is also an experiment that compares gradients in the encoder when training using LIA against training using a more typical VAE setup.

My take: The key idea behind this paper is quite promising, and I believe this paper has tremendous potential. I agree with the authors that the usefulness of implicit generative models is limited by their typical lack of an encoder network, and that existing approaches have several design drawbacks, and incorporating invertible networks and advances in flow-based modeling looks like a fruitful avenue of research.

However, I have a litany of concerns with the paper itself, concerning its high similarity with a paper published in May, its motivation, its presentation, its empirical evaluation, and the analysis presented within. While my concerns and suggestions are extensive, this paper is perhaps unusual in that all of the issues I have are quite fixable; the core idea is good, but its realization and presentation in the paper need a substantial amount of revision. I am currently giving this paper a weak reject as I do not believe it is ready for publication, but I believe that with an overhaul this paper could be a more clear accept.

Update, post rebuttal:

Thanks to the authors for their response. While I appreciate their insistence that my issues with the paper likely stem from my simply not understanding it (or the underlying topics), I hope they can appreciate that such an appeal is unlikely to allay said concerns. Pointwise:

1. While there is of course a difference between the LIA setup and the GLF setup, regardless of the two-stage training or the inclusion of an adversarial or perceptual loss or any other bells and whistles that get attached, the fact remains that the resulting architecture for both LIA and GLF is an autoencoder with an invertible model in the middle. Arguing that the LIA setup is somehow fundamentally different is akin to arguing that a VAE-GAN is an utterly different model from a VAE with a VGG perceptual loss. Yes, they're optimizing slightly different things (distribution-wise differences vs sample-wise differences) , they have different characteristics, etc., but at the end of the day they're still autoencoders with extra bits on the end. The same general principle holds here. As I stated in my original review, I consider the differences relatively minor and maintain my stance that comparison is warranted.

1(a). While the authors may have completed the work over a year ago, the fact that the other work was made public multiple months before this work means that it does, in fact, count as prior work. There is plenty of precedent at ICLR for work which appeared on arXiv well before submission date to be considered as prior work. I understand that this can be frustrating if the authors have previously submitted to other conferences and wished to wait until acceptance before making the work public, but that is a personal choice that does not change the nature of the situation.

2. If interfacing with image manipulation techniques is the motivation for improving reconstructions, this motivation should be clearly stated in the paper. After rebuttal there is still no mention of this motivation, which suggests to me that the authors expect all readers to consider "reconstruction" (which I again posit is not really a task) to matter intrinsically.

3. I once again appreciate that the authors hold that this reviewer is incorrect about basic facts. The forward KL placed on the latent space of a VAE only encourages it to resemble a particular distribution (typically isotropic gaussian) but the information content passed through the bottleneck can indeed grow with the size of the latent space, as one can see experimentally by ablating the latent dimensionality. This general principle should also be reinforced when one considers that flow models with exact inference (i.e. perfect reconstructions) require dz==dx.

4. This reviewer maintains that sampling a random input from a distribution during training involves stochasticity.

5. This still does not address my concern that reconstruction is not a particularly relevant task.

7-9: Thank you for modifying the caption in this figure, though I still hold that the y-axis should be correctly labeled. This still does not address my concern that this experiment does not actually show what the authors claim it does--the magnitude of the gradient noise is not by any measure a viable indication that the inclusion of phi is doing anything meaningful in place of a standard MLP  as the comparison is instead made to an entirely different training setup.

I maintain my stance of rejection.

Original review:

First off, a paper published in May on arXiv titled “Generative Latent Flow” (GLF) proposes an idea which is in essence identical to the one proposed in this paper. In GLF, a VAE is trained, but rather than using the reparameterization trick, a true normalizing flow is trained to model the distribution of the output of the encoder (i.e. with NLL using the change of variables formula common to flow-based models), such that the training of the actual autoencoder is truly deterministic (in the sense that at no point is an epsilon~p(z) sampled like in a normal VAE. The core difference between LIA and GLF is that GLF learns to model the distribution of the encoder outputs to enable sampling, while LIA incorporates an invertible model into a generator which explicitly through sampling, and then fits an encoder post-hoc. There are other differences in implementation and the choice of datasets, but those are (IMO) minor details relative to the core similarity. Given that GLF was published 4 months before the ICLR2020 deadline, this paper absolutely must be cited, compared against, and discussed. I am somewhat inclined to argue that given the similarity, LIA is merely incremental relative to GLF, but for now I think it is sufficient to point out the existence and similarity.

Second, the stated motivation in this paper is, I think, misguided. The authors argue for the need of an inference network, but they explicitly make clear that their goal is to train this network to enable reconstruction of an x given a z, rather than e.g. to learn a “good representation” (bearing in mind that what constitutes a good representation is strongly subject to debate). The authors do not provide any motivation for why reconstruction matters. At no point is an application or downstream task presented or mentioned in which good reconstructions are relevant. One might argue that choosing reconstruction quality is as arbitrary as pursuing improved sample quality (as is in vogue in GAN papers)  but there is substantial evidence that improved sample quality correlates with improved representation learning (mode dropping in GANs notwithstanding); the case is more complex for high-quality reconstructions.

Reconstruction could perhaps be motivated from the point of view of compression, but this paper makes no attempt to examine compression: rate-distortion tradeoffs are not considered, nor are any empirical metrics of compression ratio or likelihood such as bits/dim presented. Given that one can produce a model which achieves arbitrarily high-quality reconstructions by simply increasing the dimensionality of the bottleneck, I do not find reconstruction to be a compelling problem.

One might also argue that improved reconstruction capacity is indicative of better ability to fit the distribution (i.e. less mode dropping), but in the LIA setup the generator is trained as a standard StyleGAN with the only modification being the replacement of the MLP with Phi, so there’s no reason to believe that the implicit model defined by G has been meaningfully affected by the inclusion of the post-hoc trained encoder.

If the authors wish to pursue “reconstruction” as the primary motivation for learning an encoder, I would suggest they spend more time discussing compression and the latent bottleneck, as well as performing more detailed empirical evaluations (explained below). Basically, *why* does reconstruction matter? Alternatively, the authors could demonstrate the usefulness of their learned encoders for downstream tasks to indicate that the representations they learn are of high quality and useful.

Third, the presentation of this paper needs a lot of work. There are typos and confusing statements throughout, as well as several instances of overstatement.

The key insight of this paper appears to be that “having an invertible network at the input to the generator makes it more amenable to post-hoc learning an encoder.” If I understand correctly, the only difference between this method and Encoded StyleGAN is that this paper uses an invertible model in place of the StyleGAN MLP. If this is the case, then the paper needs to (a) make clear the minimality of this difference and (b) devote substantial exposition to exploring the difference and why this is important (see my comments in the experimental section).

Phrases like “the two-stage training successfully handles the existing issues of generative models” suggests that this method has solved all of the problems in generative modeling, which the authors have by no means demonstrated to be the case.

Calling the two stage training “Stochasticity free” is incorrect—if you’re training the model as a GAN, then (1) you’ll be sampling z’s in the first place so it already has a latent distribution defined and (2) the end result of training will be much more variable than, say, training with a likelihood metric. There is a *ton* of stochasticity in the first stage of training!

The paper states several times that the “critical limitation” of adversarial models is their lack of an encoder. While implicit generative models do not generally require an encoder, there are plenty of methods (BiGAN by Donahue and ALI by DuMoulin, along with all the VAE-GAN hybrids) that jointly learn encoders, and much work on training an encoder post-hoc. These methods are acknowledged in the related work, but I think they should be taken into consideration when describing this “critical limitation.” While not having an encoder does indeed hinder or prevent the use of an implicit model for inference, I think stability, mode dropping, and mode collapse are more prominent issues with GANs. I think the authors might do better to say something to indicate that the challenge is to train a model which both has sharp, high-quality samples (as with GANs) which is still capable of inference or explicit likelihood evaluation (VAEs, etc).

In general, I found the description of the model itself to be confusing, and needed several thorough read-throughs just to understand what was going on: what was being frozen when, the fact that the model is just a GAN with a post-hoc trained encoder--I felt that there was a lot of obfuscatory language obscuring the actual simplicity of the method (which might arguably be its strength).

While I would generally like the paper’s exposition to be improved, I understand that saying “write it better” is unhelpful so please see my individual notes at the end of this review for additional specific points.

Fourth, I found the empirical evaluations to be somewhat weak. To be clear, the results appear to be very good-the model retains the sample quality of  StyleGAN (at least as far as can be seen from the presented samples) while achieving noticeably higher-quality reconstructions on all the tested datasets. However, the metrics used for evaluation are limited—while at least MSE is presented, I would again stress that reconstruction is an odd metric to use when other factors like compression rates are not considered. While it is interesting to note that in this exact setup (mostly dim_z=512) LIA outperforms the baselines wrt the chosen metrics, a more thorough evaluation would, for instance, sweep the choice of dim_z, and ideally present NLL results (which I think are possible to evaluate given that LIA has a flow model even if it’s not trained using NLL, but I’m not 100% sure on this front and am open to counterarguments on this front).

What’s more, the datasets chosen are all single-class datasets with a massive amount of data—as far as generative modeling is concerned, these are very datasets with a minimal amount of variation. This is critical because the LIA method relies on pre-training a GAN, meaning that it does nothing to deal with problems like mode dropping and mode collapse. While we may not see much mode dropping on these very easy datasets (where there are, essentially, very few modes), this is still a substantial problem in the general case, as can be seen by results on e.g. ImageNet. If your GAN collapses or drops modes then post-training the encoder is not likely to be able to recover them. This is also arguably a weakness of this paper relative to GLF which incorporates the encoder into the training loop of the decoder and is likely to be better at covering modes.

Accordingly, I have substantial concerns that this method will not work well on datasets outside of these highly-constrained, nearly-unimodal, single-object, very-high-data datasets. While I would of course prefer to see results on something massively multimodal like ImageNet (training on a 100-class subset @ 64x64 resolution would be about 100,000 images and should be even less hardware intensive than the already performed experiments) I am aware of how cliché it is for reviewers to ask for imagenet results. Auxiliary experiments on CIFAR-100 or something with more than one class would go a long way towards allaying my concerns on this front.

Next, no error bars are presented; this is simply inexcusable. Given that no hardware requirements are presented it is difficult to judge if expecting multiple runs is unreasonable but unless each run requires weeks of the authors’ full hardware capacity, there is no reason for the authors not to include error bars or expected variances on the numbers for as many of their experiments as possible.

Further, I found the experiment in 5.3 to be confusing and the results irrelevant to the argument made by the authors. First of all, what does it mean that the “gradients” are plotted in the figures relating to this experiment? Are these gradient norms for a layer in the network, and if so, what type? Is the loss in Figure 5c the training loss or the test loss? I also disagree that the VAE “gradients” are “more unstable” than the LIA “gradients,” they are simply noisier. I do not see why the increased gradient noise relative to LIA is indicative of the superiority of the method, but is instead entirely expected given that noise is explicitly injected into a standard VAE—I would argue that the change in gradient noise is simply the result of removing the stochasticity, but it says nothing as to whether or not the LIA method is better than the VAE method. Again, I agree that using an invertible network in some capacity is preferable to using the reparameterization trick, but I found this specific experiment to be distracting.

I think the paper would do better to explore the importance of the invertible network relative to the exact same procedure but with the invertible network replaced with an arbitrary MLP of similar capacity. This appears to be what the encoded styleGAN model is, but I think it would do more to elucidate the key insights of this paper if the analysis was to focus more on this front. Why is it helpful to have an invertible phi in place of the StyleGAN MLP? What happens as the capacity of this portion of the model is increased or decreased? What is the form of the distribution output by Phi (maybe just some histogram visualizations along different dims?), and how does it compare to that of the typical MLP? What is the form of the distribution output by the encoder, and how does it differ from (a) the analytical latent distribution in the case of encoded styleGAN and (b) the empirical latent distribution of LIA? There’s quite a bit to explore there but this paper doesn’t dig very deep on this topic.

I recognize that the amount of suggestions and changes I have listed are exceptionally large (more than I’ve personally ever written before, for sure), and I want to make it clear that I don’t expect the authors to address them all in the limited timespan of the rebuttal period. While this unfortunately may mean that there is simply not enough time for my concerns to be addressed, if this is the case then I hope these suggestions prove useful for publication in the next conference cycle, where this paper could be very strong. As it is, given the extent of my concerns, this paper is currently sitting at about a 4/10 in my mind.

Minor notes:

“In the parlance of probability,” page 2. I liked this alliteration a lot. This paragraph as a whole was quite clear and well written.

“But it requires the dimension dx of the data space to be identical to the dimension dz of the latent space” Minor nitpick, but I would swap “dx of the data” with “dz of the latent space” in this sentence, to make it clear that the model’s latent dimensionality is constrained by the dimensionality of the data. As written it makes it sound like it’s the other way around.

“The prior distribution can be exactly fitted from an unfolded feature space.” While flows have exact inference, saying that you can exactly fit the distribution of the encoder is arguably inaccurate unless you can show perfect generalization. Specifically, if you attain 0 training loss for the flow, do you also have 0 test loss (i.e. the NLL of the flow on  the encoder’s output for test samples is also minimized).

Furthermore, the phrasing “unfolded feature space” (used elsewhere in the paper) is confusing and not in common usage—does this mean the output of the encoder, or some sort of Taylor expansion? It’s not immediately clear, and I would recommend the authors find a different way to express what they mean.

“Therefore the training is deterministic” Training is not deterministic if the first stage of training involves training a GAN. You are still sampling from a preselected prior in this stage.

“As shown in Figure 1f, we symmetrically embed an invertible neural network in the latent space of VAE, following the diagram of mapping process as…” This sentence is confusing. The term “embed” has a specific meaning in the literature: you might use word embeddings, or embed a sample in a space, but to “embed a [model] in a latent space” doesn’t make sense to me. I think the authors would do well to use more standard terminology, and to reconsider their description of the model to be more concise and clear.

“Our primal goal is to faithfully reconstruct real images from the latent code.” Primal should be primary. I would also like to see this motivated better—why do you care to exactly reconstruct real images? Is there a downstream task where this is relevant or an intrinsic reason why we should care about being able to attain exact reconstructions?

“indispensable discriminator.” Indispensible means “something you can’t get rid of,” whereas it would appear the discriminator is not used after  training (and is frozen after the first stage)—do the authors perhaps mean “dispensable” or “disposable”?

“The interesting phenomenon is that the StyleGAN with encoder only does not succeed in recovering the target faces using the same training strategy as LIA, even though it is capable of generating photo-realistic faces in high quality due to the StyleGAN generator” This sentence is confusingly written and poorly supported. While I do agree that the LIA reconstructions are superior to the encoded styleGAN reconstructions, exactly what do the authors mean that LIA “recovers” the target faces while StyleGAN does not? The LIA reconstructions are not identity preserving—while most of the semantic features are the same, and the model does do a good job of picking up on unusual details such as hats, the facial identities are definitely not preserved (i.e. for every face in row 1 and row 2, I would say that the two faces belong to different people with similar features, but they are still definitely different people) .

“This indicates that the invertible network plays the crucial role to make the LIA work” This statement is unsupported. There are a number of differences in training setup, and the authors, in this reviewer’s opinion, have not presented evidence to indicate that the use of the flow model is specifically responsible for this. Specifically, what would happen if during the decoder training stage, the invertible network was not employed? While I do believe that the inclusion of the invertible network is important, the authors should go to greater lengths to elucidate exactly what it does (see my comments above in the experimental section re. the shape of the distribution and how the encoder ends up matched to the decoder depending on what the actual latent distribution is from the POV of the generator).

“To further evaluate LIA on the data with large variations” The choice of three single-category, single-subject datasets for evaluation is strictly at odds with this statement. These are highly constrained, clean datasets with tremendous amounts of data per class, which are substantially less difficult to model than e.g. ImageNet

“They will be made them available for evaluation.” -> “These subsets will be made available for evaluation”

“The experimental results on FFHQ and LSUN databases verify that the symmetric design of the invertible network and the two-stage training successfully handles the existing issues of generative models.” This statement is far too strong—saying a method “successfully handles the existing issues of generative models” suggests that this method is the end-all be-all and has solved the problem of generative modeling entirely. I would suggest the authors dial back the strength of this claim.

“Table 2: FID accuracy of generative results.” What is FID accuracy? Do the authors just mean FID?

Specify hardware used and training times, at least in the appendix.


**Experience Assessment:**

I have published in this field for several years.

**Review Assessment: Checking Correctness Of Derivations And Theory:**

I carefully checked the derivations and theory.

**Review Assessment: Checking Correctness Of Experiments:**

I carefully checked the experiments.

**Review Assessment: Thoroughness In Paper Reading:**

I read the paper thoroughly.

---

> ### Author Response · Authors · 2019-11-10
> **To Reviewer #1**
>
> A1:  “I have a litany of concerns with the paper itself, concerning its high similarity with a paper published in May, its motivation, its presentation, its empirical evaluation, and the analysis presented within.”,  “...titled “Generative Latent Flow” (GLF) proposes an idea which is in essence identical to the one proposed in this paper.”
> Q1: First, saying the idea of GLF “is in essence identical to the one proposed in this paper” is incorrect. 1) About the principle. To improve the variational inference, the nature of GLF is to replace KL-divergence in VAE with the log-likelihood of normalizing flow. However, we did not use any optimization about normalizing flow. We only use the invertibility of the invertible network to establish the bijective mapping between disentangled features w and the associated latent code z in the latent space. In fact, there is no variational inference involved in our algorithm. 2) About the architecture. The algorithmic architectures between GLF and our LIA are totally different. You may know this by examining Figure 1 in the GLF paper and Figure 1 in our paper.
>
> Second, using normalizing flow to improve the variational inference in VAE was first studied by Diederik Kingma much earlier in 2016 through the following paper.
>
> Improving variational inference with inverse autoregressive flow
> https://arxiv.org/abs/1606.04934
>
> The second paper about this topic is f-VAE published in 2018, which GLF is really similar to.
>
> f-VAEs: Improve VAEs with Conditional Flows
> https://arxiv.org/abs/1809.05861
>
> We have cited and discussed these two algorithms in the related work in our paper.
>
> Third, the initial version of our work was completed in November 2018. The refined version (i.e. this version for ICLR) was completed in May 2019. We can provide evidence if needed. We also checked that GLF is in submission to ICLR’20 (https://openreview.net/pdf?id=Syg7VaNYPB) as concurrent work to us.
>
> Anyway, thank you for reminding us the existence of GLF. We will cite and discuss it in the related work.
>
> A2:  “the stated motivation in this paper is, I think, misguided”, “The authors do not provide any motivation for why reconstruction matters”, “rather than e.g. to learn a “good representation”.
> Q2: GANs encounter difficulty when we try to get the  latent code z for a given REAL image. This is fundamental to apply GANs for real image manipulation. Many recent work achieves faithful image manipulation for synthetic images (model-generated images rather than real images), such as facial attribute manipulation [1], object adding and removal in scenes [2], and steerable image attribute manipulation [3]. When applying for a real image, a basic requirement for such tasks is that a faithful reconstruction need to be guaranteed when solving the latent code. We include more real image manipulation results in the revised version (Fig.xx) to emphasize the motivation of faithful reconstruction.
>
> [1] InterFace GAN:
> https://github.com/ShenYujun/InterFaceGAN
> [2] GAN Dissection:
> https://gandissect.csail.mit.edu/
> [3] On the "steerability" of generative adversarial networks.
>  https://arxiv.org/abs/1907.07171
>
> On the other hand, Learning a good feature for image classification is the task that has a totally different purpose from the problem we solve in our paper. Please refer to the following paper to know about this application.
>
>  Large Scale Adversarial Representation Learning
> https://arxiv.org/abs/1907.02544
>
> In this scenario, the “good” representation usually means yielding the better classification results. The reconstruction is rather flexible for such tasks. For example, it is feasible for BigBiGAN to only recover the high-level semantics of real objects rather than faithful reconstruction in our task.
>
> Q3: “Given that one can produce a model which achieves arbitrarily high-quality reconstructions by simply increasing the dimensionality of the bottleneck, I do not find reconstruction to be a compelling problem.”
> A3: This viewpoint is incorrect. For the vanilla autoencoder, we can say “high-quality reconstructions by simply increasing the dimensionality of the bottleneck”. For VAE, however, the case is totally different because there is a probability constraint on the latent space. In fact, the variational inference becomes more difficult when the dimensionality of the latent space increases due to the curse of dimensionality. This is one of the underlying reasons why learning a good VAE is so difficult, even though its architecture is simple. Please refer to the following page to understand this.
>
> Curse of dimensionality
> https://en.wikipedia.org/wiki/Curse_of_dimensionality

---

> > ### Author Response · Authors · 2019-11-10
> > **To Reviewer #1 (continued)**
> >
> > Q4: “Calling the two stage training “Stochasticity free” is incorrect—if you’re training the model as a GAN, then (1) you’ll be sampling z’s in the first place so it already has a latent distribution defined and (2) the end result of training will be much more variable than, say, training with a likelihood metric. There is a *ton* of stochasticity in the first stage of training!”
> > A4: You misunderstood our motivation and algorithm. “Stochasticity-free” refers to that there is no any stochasticity involved in the loss and optimization of our algorithm. There is NO any likelihood used in our algorithm.
> >
> > Q5: “However, the metrics used for evaluation are limited—while at least MSE is presented, I would again stress that reconstruction is an odd metric to use when other factors like compression rates are not considered.”
> > A5: We provided three metrics for the evaluation, including MSE. And our work has little relevance with compression.
> >
> > Q6: “What’s more, the datasets chosen are all single-class datasets with a massive amount of data”, “While I would of course prefer to see results on something massively multimodal like ImageNet”
> > A6: The datasets we used are the same as the ones used by the state-the-art unconditional generative models like ProGAN (Karras et al., ICLR’18) and StyleGAN (Karras et al., CVPR’19). Actually, we follow the experimental setup of StyleGAN.
> >
> > Q7: “Further, I found the experiment in 5.3 to be confusing and the results irrelevant to the argument made by the authors.”, “I do not see why the increased gradient noise relative to LIA is indicative of the superiority of the method.”
> > A7: The experiment is to show the superiority of using the invertible network. We explain this in detail in section 5.3.
> >
> > Q8: “Why is it helpful to have an invertible phi in place of the StyleGAN MLP?”
> > A8: We explain it in section 2.1, section 3, and section 5.3.
> >
> > Q9: “what does it mean that the “gradients” are plotted in the figures relating to this experiment?”
> > A9: It is the norms of gradients. We make it clear in the revised version.

---

### Official Review · AnonReviewer2 · 2019-10-26
**Official Blind Review #2**

**Rating:** 3

**Review:**

This paper develops a new generative model called latently invertible autodecoder (LIA). The major aim is to conduct variational inference for VAE and encoding real-world samples for GAN. My understanding is that the authors tried to achieve this by decomposing the framework into a wasserstein GAN and a standard autoencoder. I believe this paper contains promising idea.

The experiment is very thorough, and the results show that LIA achieves good empirical performance.

The method is not presented in a user-friendly fashion, and the presentation can be improved.

I must admit that I do not work on this field, and cannot judge this paper with more details.

**Experience Assessment:**

I do not know much about this area.

**Review Assessment: Checking Correctness Of Derivations And Theory:**

I did not assess the derivations or theory.

**Review Assessment: Checking Correctness Of Experiments:**

I did not assess the experiments.

**Review Assessment: Thoroughness In Paper Reading:**

I read the paper at least twice and used my best judgement in assessing the paper.

---

### Official Review · AnonReviewer3 · 2019-11-02
**Official Blind Review #3**

**Rating:** 3

**Review:**

The work in the paper is interesting and seems to show some empirical progress on image generation. However, the work in this paper can be seen as adding the losses from GANs and VAEs together to help learn a better generative model, which is not very novel. The invertible part is help for training as well. Still more detail about how the method works would be very helpful. While there is lots of math in the paper it is difficult for the reader to follow and often not well motivated why these choices are made. For example, the optimization is split into two different updates steps because they can't be combined mathematically. Yet, performing two different types of update steps can often favour whichever is easier and the other is noise. More details on how this process was made successful are important to discuss in the paper.


More detailed comments:
-	The conjecture that VAEs produce blurry images needs a reference.
-	I am not sure if GANs are limited by the lack of an encoder. It could be that the introduction of an encoder is exactly what makes it difficult for VAE to learn complex and detailed image generation.
-	The second claim in the paper (not affected by posterior collapse) should be proved in the paper or at least illustrated in some way. Currently, this claim is not well backed up in the paper.
-	THe claim that the method will have linear interpolation and vector arithmetic will need to be more rigorously proven. Right now it seems a little too much like proof by picture.
-	I do like figure 1 and 2. They help explain the method rather well.
-	In the begining of the experiment section, there are a number of hyperparameter values defined yet what these hyper parameters are is not explained. More detail needs to be added in this area for the reader to understand the experiments.
-	The latent encoding size use is rather large.


**Experience Assessment:**

I have read many papers in this area.

**Review Assessment: Checking Correctness Of Derivations And Theory:**

I assessed the sensibility of the derivations and theory.

**Review Assessment: Checking Correctness Of Experiments:**

I assessed the sensibility of the experiments.

**Review Assessment: Thoroughness In Paper Reading:**

I read the paper at least twice and used my best judgement in assessing the paper.

---

> ### Author Response · Authors · 2019-11-10
> **To Reviewer #3**
>
> Q1:  “However, the work in this paper can be seen as adding the losses from GANs and VAEs together to help learn a better generative model, which is not very novel.”
> A1:  This is a misunderstanding on our algorithm. We do not use the loss of KL-divergence in VAE. Our algorithm is not the loss combination of GANs and VAEs either.
>
> Q2: “While there is lots of math in the paper it is difficult for the reader to follow and often not well motivated why these choices are made.”
> A2: We only used the very basic mathematical expressions to make writing more accurate. In order to help understanding, we plotted Figures 1 and 2 to explain the principle, architecture, and training strategy of our algorithm. We also used equation (2) to show the mapping details of our algorithm.
>
> Q3:  “The conjecture that VAEs produce blurry images needs a reference.”
> A3: The following papers clearly show the blurry images generated by VAEs, we will include them in the revised version. In our experimental section, we have also shown the blurry images generated by VAEs.
>
> Autoencoding beyond pixels using a learned similarity metric
> https://arxiv.org/abs/1512.09300
> Wasserstein Auto-Encoders
> https://arxiv.org/abs/1711.01558
> Implicit Discriminator in Variational Autoencoder
> https://arxiv.org/abs/1909.13062
>
> Q4: “I am not sure if GANs are limited by the lack of an encoder.”
> A4: Lacking encoder is one of the fundamental problems for GANs. It greatly limits the GANs for real image applications such as facial image manipulation [1] and semantic scene editing [2]. If we want to edit any given real images with GANs, we need to know the corresponding latent code z. This cannot be achieved with the vanilla GAN, thus posing the problem.
>
> [1] InterFace GAN
> https://github.com/ShenYujun/InterFaceGAN
> [2] GAN Dissection
> https://gandissect.csail.mit.edu/
>
> Q5: “The second claim in the paper (not affected by posterior collapse) should be proved in the paper or at least illustrated in some way.”
> A5: We didn’t use any loss or optimization about posterior probability. So, our algorithm is totally free from this problem.
>
> Q6: “THe claim that the method will have linear interpolation and vector arithmetic will need to be more rigorously proven.”
> A5: To the best of our knowledge, there is no work that can rigorously prove linear interpolation and vector arithmetic about GANs up to now. We follow the convention of the baseline DCGAN and the state-the-art StyleGAN to illustrate these aspects by experiments. All the papers about GANs on this aspect perform this task as the same as ours.
>
> Q7: “there are a number of hyperparameter values defined yet what these hyper parameters are is not explained. More detail needs to be added in this area for the reader to understand the experiments.”
> A7: We explain these hyperparameters in the main context. We will explain them again in the experiment for easy understanding according to your advice. Thanks for the reminder.
>
> Q8: “The latent encoding size use is rather large.”
> A8: The 512-dimensional latent codes are usually applied in GAN works such as  ProGAN (Karras et al., ICLR’18) and StyleGAN (Karras et al., CVPR’19). We follow the convention.

---

> > ### Comment · AnonReviewer3 · 2019-11-13
> > **Thank you for your clarifications**
> >
> > Thank you for your clarifications
> >
> > These comments have helped clear up my understanding of some important details.

---

### Official Review · AnonReviewer4 · 2019-11-04
**Official Blind Review #4**

**Rating:** 3

**Review:**

Summary:

the authors of this paper propose a two-stage model consisting of a Wasserstein GAN and an autoencoder. The goal is to learn an encoding of data within the GAN framework.
The idea is inspired by the concept of VAEs. However, instead of maximising the ELBO, the authors propose to learn/represent the generative model by a Wasserstein GAN (first stage). Here, the architecture is crucial: an invertible MLP_1 is used to map from a standard normal prior into the feature space; then, a classical MLP_2 maps into the data space
In the second stage, MLP_2 serves as decoder to train an encoder. By combining the latter with MLP_1, data can be encoded into the latent space.

The authors experimentally show that their method leads to improved reconstructions compared to previous GAN-based methods.


In the following, a few concerns:

1) The authors motivate their approach with the goal of "encoding real-world samples" without accepting disadvantages of VAEs such as "imprecise inference" or " posterior collapse". However, the comparison to VAEs is difficult since the latent representation of the data learned by VAEs differs from the one of LIA.
Fore example, in contrast to VAEs, where similar data is clustered in the latent space, this is not necessarily the case for GANs (e.g. Mukherjee et al., 2019). Experiments regarding the learned latent representation are missing in the paper (the interpolation experiment in the appendix might be a starting point).

2) The authors use posterior collapse in VAEs as a main argument for introducing LIA. However, it is easy to avoid as stated in e.g. Bowman et al. (2015) or Sønderby et al. (2016), and hence this argument doesn't make a strong case for LIA.

3) It is difficult to interpret the experiments in Fig. 5: the first 10 iterations might not be very significant.

4) Experimental details are missing. I would appreciate to have model architectures in the appendix (even if the authors are going to make the source code publicly available).

5) How were the accuracies of the generations in Tab. 2 computed?



**Experience Assessment:**

I have published one or two papers in this area.

**Review Assessment: Checking Correctness Of Derivations And Theory:**

I carefully checked the derivations and theory.

**Review Assessment: Checking Correctness Of Experiments:**

I carefully checked the experiments.

**Review Assessment: Thoroughness In Paper Reading:**

I read the paper thoroughly.

---

> ### Author Response · Authors · 2019-11-10
> **To Reviewer #4**
>
> Q1:  “However, the comparison to VAEs is difficult since the latent representation of the data learned by VAEs differs from the one of LIA.”
> A1: Indeed comparing the latent representations of the VAEs and LIA directly is difficult, but we have compared the quality of the reconstruction images by the two methods in the experiment and our method achieved much better reconstruction accuracy.
>
> Q2: “Experiments regarding the learned latent representation are missing in the paper (the interpolation experiment in the appendix might be a starting point)”
> A2: We add more experiments on interpolation and attribute manipulation, as well as the analysis on entanglement of latent representations in the revised version.
>
> Q3: “The authors use posterior collapse in VAEs as a main argument for introducing LIA. However, it is easy to avoid as stated in e.g. Bowman et al. (2015) or Sønderby et al. (2016), and hence this argument doesn't make a strong case for LIA. ”
> A4: The posterior collapse problem of VAE with deep convolutional neural networks still exists for the synthesis of complex data like real scenes and objects, especially for high-resolution images. This is one of the reasons why state-of-the-art VAEs cannot synthesize images as good as the GANs.
>
> Q4: “It is difficult to interpret the experiments in Fig. 5: the first 10 iterations might not be very significant.”
> A4: The x-axis is re-scaled by 10,000. We didn’t explain it in the caption. We update it in the revised version.
>
> Q5: “Experimental details are missing. I would appreciate to have model architectures in the appendix.”
> A5: More experimental details and model architectures are provided in the revised version, the model architectures are attached in Appendix.9 in the revised version.
>
> Q6:  “How were the accuracies of the generations in Tab. 2 computed?”
> A6: We followed the standard evaluation in StyleGAN paper (Karras et al., CVPR’19) to compute FIDs in Table 2. We randomly generated 50,000 fake images and randomly sampled 50,000 real images. Then we computed FIDs with these two datasets.

---

### Decision · Program_Chairs · 2019-12-19

**Decision:**

Reject

**Comment:**

A nice idea: the latent prior is replaced by a GAN.  A general agreement between all four reviewers to reject the submission, based on a not thorough enough description of the approach, and possibly not being novel.